Detection of terrestrial mammals using environmental DNA during heavy rainfall events and associated influencing factors

Xu Chen 1
Nukazawa Kei nukazawa.kei.b3@cc.miyazaki-u.ac.jp 2
1 Interdisciplinary Graduate School of Agriculture and Engineering, University of Miyazaki , Miyazaki , Japan
2 Department of Civil and Environmental Engineering, Faculty of Engineering, University of Miyazaki , Miyazaki , Japan
Brannelly Laura
Electronic publication date: 2025 Oct 15
Publication date: 2025
Volume: 13
Electronic Location ID: e20166
Received 2025 Jun 25; Accepted 2025 Sep 11
Copyright: ©2025 Xu and Nukazawa
Copyright year: 2025
Copyright holder: Xu and Nukazawa
License: This is an open access article distributed under the terms of the Creative Commons Attribution License, which permits unrestricted use, distribution, reproduction and adaptation in any medium and for any purpose provided that it is properly attributed. For attribution, the original author(s), title, publication source (PeerJ) and either DOI or URL of the article must be cited.
License URL: https://creativecommons.org/licenses/by/4.0/

Keywords: Environmental DNA, Terrestrial mammals, Turbid water, Pore size, Digital PCR, Generalized linear mixed model

Funding: The River Fund of The River Foundation, Japan, Japan Society for the Promotion of Science 24KK0086 24H00329 25K00041 The Kurita Water and Environment Foundation This study was supported by the River Fund of The River Foundation, Japan, Japan Society for the Promotion of Science (Award Number: 24KK0086, 24H00329 and 25K00041) and the Kurita Water and Environment Foundation. The funders had no role in study design, data collection and analysis, decision to publish, or preparation of the manuscript.

==============================
Recent developments in environmental DNA (eDNA) analyses have facilitated non-invasive and cost-effective ecological monitoring. Based on eDNA of terrestrial species released into water upon contact, simultaneous detection of aquatic and terrestrial species is feasible. However, an efficient sampling design for terrestrial vertebrate eDNA in aquatic environments has not yet been established because DNA is rarely released into these environments. In this study, we targeted eDNA transported from land to rivers through surface runoff during rainfall in three rivers and one irrigation channel within the Kiyotake River system, Japan. We quantified the eDNA concentration of a specific terrestrial vertebrate (Bos taurus) using digital polymerase chain reaction (PCR) and examined the efficiency of using filter papers with different pore sizes (0.7 µm and 2.7 µm). We also assessed the influence of various environmental factors (e.g., rainfall characteristics described by the parameters of Gaussian distribution, water turbidity) on eDNA detection across different rainfall events. During the surveys, target DNA was detected in 42 out of 47 samples, suggesting the feasibility of consistently detecting terrestrial mammals from stormwater runoff. Overall, compared with the glass fiber filter with larger pore size, the smaller pore size filter captured more eDNA. The generalized linear mixed model revealed that prolonged rainfall duration, turbidity, and pH had a significant positive effect on eDNA concentration, whereas the distance from the assumed point of entry into the river to the sampling point had a significant negative effect. These results suggest that the runoff and transport of eDNA from terrestrial areas to rivers are enhanced under prolonged rainfall conditions, although eDNA degrades while transported along a longer watercourse by biochemical decomposition and sedimentation.

Introduction

Biological surveillance using environmental DNA (eDNA) has rapidly developed in the last two decades (Ficetola et al., 2008; Minamoto et al., 2012; Pawlowski et al., 2021; Takahashi et al., 2023). Environmental DNA refers to the DNA present in environmental samples (e.g., water and soil) via the introduction of biological material shed from an organism (e.g., hair, skin cells, fecal matter) (Taberlet et al., 2012a). Based on the detectability of target DNA, the presence or absence of species can be determined. It has been reported that, compared with traditional surveys such as direct capture and visual detection, eDNA methods have more advantages, including low cost, rapidity, and non-invasiveness (Dyson et al., 2024; Fediajevaite et al., 2021; Suren, Burdon & Wilkinson, 2024).

Environmental DNA analysis is widely used in surveys of aquatic species from rivers, lakes, and sea (Goldberg et al., 2011; Mächler et al., 2014; Minamoto et al., 2017; Reinhardt et al., 2019; Takahara, Minamoto & Doi, 2015; Valentini et al., 2016). In recent years, this method has gradually been extended to terrestrial ecosystems. Studies have shown that eDNA extracted from various terrestrial substrates—such as the surfaces of flowers, soil, and air—can be used to investigate the diversity of terrestrial vertebrates (Calvignac-Spencer et al., 2013; Clare et al., 2021; Lynggaard et al., 2024; Taberlet et al., 2012b; Thomsen & Sigsgaard, 2019). On the other hand, when terrestrial mammals come into contact with water bodies (e.g., bathing, defecating, or drinking), their DNA will be released into the water and can be detected through water samples (Rodgers & Mock, 2015; Ushio et al., 2017). Based on this principle, another study detected aquatic, semi-aquatic, and terrestrial mammals simultaneously from Amazon River water samples using eDNA (Coutant et al., 2021). In addition to mammals and fish, other vertebrate taxa (birds, amphibians, and reptiles) have been also detected from water bodies (Nordstrom et al., 2022; Ritter et al., 2022). These studies suggest that eDNA in water bodies has the potential to simultaneously assess both aquatic and terrestrial species diversity at broader spatial scales (Macher et al., 2021; Mariani et al., 2021). However, for species that rarely come into contact with water or have low abundance, eDNA released into rivers may not be detected even if they are present around the sampling location (Harper et al., 2019). In addition, the accelerated degradation of eDNA in water, depending on environmental factors such as pH, water temperature, and flow distance, makes detection even more difficult (Mauvisseau et al., 2022; Nukazawa, Hamasuna & Suzuki, 2018; Seymour et al., 2018; Strickler, Fremier & Goldberg, 2015).

Previous studies speculated that surface runoff during a rainfall event may transport terrestrial eDNA to rivers and other water bodies (Coutant et al., 2021; Yang et al., 2021), although the idea has not yet been verified through a systematic study design. A study on the effect of rainfall on eDNA in vegetation found that rainfall rapidly removed eDNA from the surface of the plants, regardless of whether the vegetation surface was smooth or rough (Valentin et al., 2021). These results imply that, through a physical process, terrestrial eDNA is transported to rivers through surface runoff associated with rainfall. Therefore, eDNA surveys of rivers during rainfall events may allow for the rapid detection of terrestrial vertebrates that would normally have little contact with water bodies. However, it must be considered that surface runoff also carries fine to coarse sediments and transports them into the water bodies, resulting in high water turbidity. Therefore, a specific method is needed to efficiently recover eDNA from highly turbid water samples, which likely contain PCR inhibitors.

Many studies have been conducted in turbid environments utilizing eDNA sampling (Holmes et al., 2024; Ip et al., 2021; Kumar et al., 2022a). When filtering turbid water, filter clogging frequently occurs, which can hinder the collection of eDNA. Therefore, filter pore size cannot be generalized across all studies (Kumar et al., 2022b). For some species or environments (including waterbodies with high primary productivity), larger pore sizes are preferred (Jo et al., 2020). To prevent clogging, some studies have attempted to decrease the filtration volume, but this has resulted in a lower detectability of the target species (Deiner et al., 2015; Li et al., 2018). Environmental DNA was also detected from turbid water by concentration through centrifugation (Kusanke et al., 2020; Williams, Huyvaert & Piaggio, 2017). However, the authors sampled bathing water that contained unusually high concentrations of target eDNA. This situation differs from natural environments, where eDNA concentrations are generally low, indicating that the method may not be directly applicable under typical environmental conditions (Cooper et al., 2022). Further studies are required to better understand the impact of environmental and experimental conditions on the detectability of species in turbid water samples.

Understanding the necessary conditions for detecting eDNA from turbid water during rainfall events and establishing reliable methods could pave the way for the simultaneous monitoring of aquatic and terrestrial mammals in watershed environments. Environmental DNA metabarcoding of rainwater collected after rainfall events from tree crowns has been used to detect terrestrial invertebrate species (Macher et al., 2023). However, the effect of varying rainfall intensities on the observed species richness was not explored, and the results were barely statistically discussed. Similarly, post-rainfall eDNA surveys have been conducted in places such as rivers (Staley et al., 2018). The results of this work showed more diverse eDNA profiles compared with those during their drier sampling dates, with relatively more sequences from mammalian and bird species, which were absent during dry sampling dates. It should be noted that the study focused on the detection of fecal sources in river water; thus, the source of the detected terrestrial mammalian DNA and the environmental factors influencing the detectability remain unclear. The factors that determine eDNA detectability from stormwater should be further investigated to better understand the conditions where such a method can be applied.

Therefore, in this study, we used digital PCR (dPCR) approach to analyze eDNA of turbid water sampled during rainfall events of different magnitudes and investigated the detectability of Bos taurus, which is popularly farmed in the area. We used dPCR instead of eDNA metabarcoding because of its high specificity. As a third-generation PCR technology, dPCR allows for the absolute quantification of DNA and is less variable than measurements obtained via qPCR (Doi et al., 2015; Kuypers & Jerome, 2017; Nathan et al., 2014). The detectability of terrestrial mammals in river water during rainfall is likely to depend on the process of terrestrial eDNA transport to water and, specifically, on parameters related to degradation and sedimentation during transport and in the receiving water body. We further explored the factors influencing the detectability of terrestrial species in turbid stormwater (e.g., water quality and rainfall patterns). The study area is characterized by high annual precipitation (data: http://www.data.jma.go.jp/), which provides favorable conditions for investigating the dynamics of terrestrial eDNA under various rainfall scenarios and improving the detection rates of terrestrial species. Taking advantage of this, our study explored the variation in terrestrial eDNA during weak to heavy rainfall events, enhanced the detection efficiency of terrestrial species in turbid stormwaters, and provides a basis for optimizing relevant ecological monitoring strategies.

Materials & Methods

Study area and target species

Portions of this text were previously published as part of a preprint (https://doi.org/10.22541/au.174312906.60655689/v1). The study area included three rivers within the Kiyotake River catchment in Miyazaki Prefecture, southwestern Japan: the Mizunashi, Oka, and Tagami Rivers as well as the irrigation canal connected to the latter (Fig. 1A). The Kiyotake River is 28.8 km long and its area covers 166.4 km2 (data: https://www.pref.miyazaki.lg.jp). As land use influences the intensity of surface runoff (Guzha et al., 2018), we utilized the Advanced Land Observing Satellite (10-m accuracy, data available: https://earth.jaxa.jp/ja/data) to classify land use into 12 categories (e.g., Urban, Field, Deciduous Broadleaf Trees, Deciduous Needleleaf Trees, Fig. 1B), and further classified into four categories: forest (e.g., Deciduous Broadleaf Trees, Deciduous Needleleaf Trees), farmland (Paddy Field, and Field), urban areas (Urban), and others (e.g., Water Area, Grassland). The proportion of each land use was derived by dividing the number of meshes corresponding to the land-use type (e.g., forested area) by the total number of meshes in the watershed of the studied sites. The major watershed land used in the studied rivers are forest in the Mizunashi River (71.57%), forest and farmland in the Oka River (39.25% and 31.25%, respectively), and farmland and urban areas in the Tagami River (43.24% and 21.38%, respectively).

Figure 1 Sampling site and land use.

(A) Sampling sites, a weather station, and the target cattle shed (serving as the eDNA point source) in the Kiyotake River Basin, located in southwestern Japan, and (B) a land use distribution map. The base map was created using ArcGIS, with geographic layers obtained from the Geospatial Information Authority of Japan (https://nlftp.mlit.go.jp/ksj/). Sampling sites and source locations were plotted based on GPS coordinates. The watershed boundary was delineated using ArcGIS. All other graphic elements were prepared by the authors. Photos of the river during rainfall were taken by the authors.

Our study focused on livestock cattle species, B. taurus. As a region with a well-developed livestock industry, Miyazaki City contains multiple cattle sheds, providing widespread sources of eDNA release. Due to the small scale of each cattle shed (200∼300 m2), we hypothesize that the abundance of this model species at each site was comparatively low and that it is confined within enclosures and designated areas. Each cattle shed is situated at a certain distance and elevation from the corresponding river, making it physically impossible for the cattle to enter these natural water bodies. Therefore, the direct use of rivers by target species is negligible.

Local regulations prohibit wastewater from being directly discharged into rivers, thereby limiting eDNA from wastewater sources. The sewer system is a combined system that treats wastewater at water treatment facilities before it is discharged. This species has a low risk of false positive results due to the absence of individuals from the same species in the wild. At least one cattle shed raising individuals outdoors is present in each river catchment (Fig. 1). The specific geographic coordinates of the sampling locations were: the Mizunashi River (31.829925°N, 131.362669°E), the Oka River (31.85162°N, 131.386959°E), the Tagami River (31.838143°N, 131.417732°E), and the irrigation canal (31.839362°N, 131.414762°E) and geographic coordinates of the cattle sheds were: the Mizunashi River (31.828235° N, 131.359762°E), the Oka River (31.852260°N, 131.385643°E) the Tagami River and the irrigation canal (31.839116°N, 131.413398°E).

Sampling

A scheme of field investigations and experiments is presented in Fig. 2. Prior to water sampling, one and 0.25 L plastic bottles, tethered buckets, cooler boxes, and a scoop were sterilized with 10% bleach for at least 30 min to prevent contamination by DNA and microorganisms other than the target species. We collected samples multiple times after and during moderate to heavy rainfall events from 2021 to 2023. Specifically, we collected samples twice a day (Sampling 1 and 2) on September 14, 2021, November 22, 2021, May 12, 2022, and June 2, 2023, to understand the temporal pattern of eDNA detection. The time interval between sampling was approximately 2 h. In addition to these dates, samples were collected once a day, with the sampling on November 22, 2021, performed after rainfall. Each sampling involved collecting four L of river water (1 L × 4 replicates, total of four L) for filtration and another 0.25 L for the measurement of basic environmental parameters, i.e., pH, electrical conductivity (EC), and turbidity. After securely tightening the bottle caps, the samples were placed into 45 L transparent plastic bags and then stored in a cooler box to isolate them from samples collected at other locations. Water temperature was measured at sampling sites using a bar-shaped mercury thermometer, and the other parameters were measured from 0.25 L samples transported to the laboratory. EC and pH were measured using a portable multiwater quality meter (HORIBA TOA DKK, Kyoto, Japan) and turbidity was measured using a turbidity meter (PT-200_TROAM, Asahi Science, Tokyo, Japan) in the laboratory. Turbidity, EC, pH, and temperature were not measured in September 2021; temperature was not measured in November 2021; and EC was not measured in June 2023 due to technical problems. During the surveys, tethered buckets were used to collect water samples in cases where entering the river was physically impossible. Each sampling event was completed within 2 h. To prevent sample-to-sample contamination and contamination from external sources during the survey and experiments, a one L of sterile distilled water prepared in the laboratory was placed in a cooler box and transported to the sampling site. After being opened and closed at the sampling site, it was returned to the laboratory with river water samples. We obtained rainfall data for the 24-h period before each sampling event under rainfall conditions from the nearest meteorological station (Akae Observatory, https://www.data.jma.go.jp/obd/stats/etrn/index), and the data are shown in Appendix 1. As a negative control, one L of surface water was collected following a period without rainfall at the same survey sites on December 27, 2021, and November 28, 2022. In addition to one L of surface water, riverbed sediment was collected on August 8 and December 13, 2024.

Figure 2 Flow diagram of the sampling process and experiment.

The water quality parameters examined were pH, electrical conductivity, water temperature, and turbidity. During rainfall events, four L river water was collected (1 L × 4 replicates), and filtered using GF/F and GF/D filters (two replicates each). During non-rainfall events, one L river water and riverbed sediment were collected. For the sediment samples, two eDNA extraction methods were employed.

Filtration and extraction

Figure 2 shows the experimental process. The collected samples were immediately transferred to the laboratory in a cooler box with ice packs, and filtration was completed within 5 h using a glass fiber filter with a pore size of 0.7 µm (GF/F; GE Healthcare Japan, Tokyo; referred to as “small filter”). To avoid filter paper clogging when filtering turbid water, we also used a filter membrane with a pore size of 2.7 µm (GF/D; GE Healthcare Japan, Tokyo; referred to as “large filter”), which is the largest pore size among the glass fiber filters. We compared the filtration efficiency and eDNA recovery rate from each filter with two replicates per filter. The samples collected without rainfall were filtered using only GF/F (one replicate). The funnels and bases (Advantec, Taipei, Taiwan) used for filtration were sterilized in advance via autoclaving (LSX-700, TOMY). Before processing the samples, the funnels, bases, clamps, and tweezers were sterilized again using 10% bleach. We shook the bottle before filtering to prevent eDNA from settling at the bottom due to prolonged filtration time. When the filtration rate dropped below 20 drops/10 s, the filter was replaced with a new one, with a maximum of two filters used per sample. Any remaining unfiltered water was discarded. This limitation was intended to minimize filter clogging and reduce the labor-intensive nature of the extraction process, thereby improving the efficiency of eDNA analysis in turbid water environments (Hinlo et al., 2017). The water volume and filtration time until clogging were recorded for each filter paper. In cases where two filter papers were used for a sample, the total volume and time were used as the volume and time respectively. After processing the samples, all filter papers were frozen at −20 °C until DNA extraction.

DNA extraction was performed using the DNeasy PowerSoil Kit (Qiagen, Hilden, Germany), which produces DNA extracts free of inhibitors (Eichmiller, Miller & Sorensen, 2016). This kit is also able to capture extracellular DNA resulting from unfreezing processes (Gielings et al., 2021; Hermans, Buckley & Lear, 2018). Firstly, using sterilized scissors and tweezers, the filter papers were cut to pieces of approximately 1  × 3 mm on a clean bench that had been pre-treated with ultraviolet sterilization. The workbench was wiped with 70% ethanol between samples to prevent cross-contamination. Subsequently, DNA was extracted from the filter fragments following the kit manufacturer’s protocol and eluted finally in 100-µL volumes. After extraction, the DNA template solutions were stored at −80 °C for subsequent PCR amplification.

We adopted two methods for eDNA extraction from sediment samples. In the first approach, approximately 28 g of sediment sample was mixed with 100 mL of sterile phosphate buffer solution (PBS, pH = 7) for 2 min in a plastic bottle pre-sterilized with bleach for 30 min. Subsequently, 100 mL of the mixture was poured into a sterile funnel, taking care not to introduce sand particles, and filtered through a GF/F glass fiber filter paper (Nevers et al., 2020). In the second approach, 20 g of sediment sample was mixed with 20 mL of phosphate buffer for 15 min. Two milliliters of the mixture was centrifuged (10 min at 10,000 g), and 600 µL of the resulting supernatant was used for subsequent extraction (Rota et al., 2020; Taberlet et al., 2012b). The samples collected in Aug. 2024, and Dec. 2024 were extracted following the DNeasy PowerSoil Pro Kit manufacturer’s protocol and eluted in 100-µL volumes (since the DNeasy PowerSoil Kit has been discontinued).

eDNA quantification and PCR conditions

The species-specific cattle DNA concentration was quantified using primers and probes targeting the mitochondrial cytochrome b (Cytb) region (Dooley et al., 2004). Amplicon size is 116 bp. The primer and probe sequences were as follows: forward primer: CGGAGTAATCCTTCTGCTCACAGT, reverse primer: GGATTGCTGATAAGAGGTTGGTG, Probe: TGAGGACAAATATCATCATTCTGAGGAGCWARGTYA. The mixture was dispensed into the independent wells of a QuantStudio 3D dPCR 20-K chip using a QuantStudio 3D dPCR Chip Loader (Applied Biosystems, Foster City, CA, USA). The end-point PCR was performed using a thermal cycler (ProFlex, Applied Biosystems, Foster City, CA, USA). To determine the optimal PCR conditions, preliminary experiments were performed using a dilution series of DNA extracted from beef meat (10−2–10−3 fold, starting concentration: 4,738 copies/µL). DNA was extracted from the beef samples using the DNeasy Blood & Tissue Kit (Qiagen, Hilden, Germany) following the manufacturer’s instructions. The results obtained for the optimal PCR conditions are shown in Appendix 2. When using forward and reverse primers at 450 nM and the probe at 125 nM, an annealing temperature of 60 °C and 50 cycles yielded an adequate separation of fluorescence between positive and negative wells in the dPCR system for both 100-fold and 1,000-fold dilutions. Therefore, all samples were adopted under these conditions for the subsequent PCR amplification. The default reaction volume for QuantStudio 3D dPCR is 14.5 µL, consisting of 7.25 µL 1 × Quant Studio 3D Digital PCR Master Mix, two µL of DNA template solution, 450 nM each primer (the forward and reverse primer) and 125 nM Taqman probe. The fluorescence intensity thresholds defined in the default settings of the Quant Studio 3D Analysis Suite (Applied Biosystems, Foster City, CA, USA) were used to distinguish between positive and negative wells. However, in cases where the fluorescence intensity of positive wells was low and not clearly separated from that of negative wells in the plots, those wells were considered negative (Nukazawa, Akahoshi & Suzuki, 2020). Because the QuantStudio 3D dPCR 20-K chip has been discontinued, non-rainfall samples collected in August and December 2024 were quantified using the QIAcuity one digital system (Qiagen, Hilden, Germany), with a default reaction volume of 40 µL. The same PCR conditions as those mentioned above were used. Since the initial attempts with QIAcuity One dPCR resulted in poor filling of the Nanoplate 26K 24-well plate, the reaction volume was increased to 43 µL following the manufacturer’s recommendation, consisting of 11.25 µL Probe PCR Master Mix, two µL of DNA template solution, 450 nM each primer (the forward and reverse primer) and 125 nM Taqman probe. The mixture was dispensed into a Nanoplate 26K 24-well. Finally, the results were confirmed using the QIAcuity Software Suite. Because the Quant Studio 3D Digital PCR Master Mix and Probe PCR Master Mix may contain bovine serum components, we included two samples without DNA in each type of dPCR to check for reagent contamination (PCR control).

Environmental factors

In this study, specific environmental variables influencing the detectability of eDNA were selected, and their relationship with eDNA concentration was assessed. In the terrestrial environment, the percentages of forest, farmland, and urban areas were taken into consideration, given their relevance to the occurrence of surface runoff (Guzha et al., 2018). Furthermore, two specific distances were considered: the distance from eDNA release sources (the cattle sheds) to the study river (the shortest straight-line, referred to as land distance) and the path that terrestrial eDNA travels to reach the sampling site after being transported into the water body (nonstraight line, referred to as the waterways distance). The land and waterways distances were approximately 125 and 360 m for the Mizunashi River, 36 and 180 m for the Oka River, approximately 311 and 187 m for the Tagami River, and 17 and 134 m for its irrigation channel, respectively (Google Maps). In the aquatic environment, we considered water temperature, pH, EC, and turbidity.

The magnitude and duration of rainfall are potentially key factors regulating terrestrial eDNA runoff. To determine their influence, we fitted the rainfall time series (i.e., hyetograph) for each sampling campaign using a normally distributed probability distribution to describe the characterization of each rain event. The hourly precipitation recorded over a 24-h period at the Akae rainfall station was used as an observation value, and a function describing a calculated precipitation was developed; αf(x), where f(x) and α are the normal distribution function and a constant, respectively. Here, the time when the water sampling started was set to time 24, and 24 h before sampling started was set to time 1 to initialize the time axis; thereby, the mean (µ) of the normal distribution varied from 1 to 24. The search range for the standard deviation (σ) of the normal distribution and the constant α were set from 0 to 100 and from 0 to 1,000, respectively, because these parameters varied greatly in the preliminary attempts. Iterative parameter searches were performed using R ver. 4.3.2 (R Core Team, 2023) to determine the values of μ, σ, and α that minimized the root mean square error of observation and calculated value. A larger μ suggested that the peak rainfall occurred closer to the sampling time, whereas a larger σ indicated that similar magnitude rainfall events lasted longer (Fig. 3, Appendix 1).

Figure 3 Partial results of rainfall time-series fit with a normal distribution.

The “Sampling1” and “Sampling2” indicate the first and second sampling of the day, respectively. Sampling2 was conducted 2 h after the start of Sampling1. “σ ” and “μ” represent the variance and mean of normal distribution, here indicating duration of rainfall and intense rainfall timing, respectively. The time ‘24’ represents the sampling time for each sampling event.

Regression analysis

Generalized linear mixed models (GLMM) with the Gaussian family were used to evaluate the relationship between eDNA concentrations and environmental factors (μ, σ, turbidity, land distance, waterways distance, percentage of each land use, pH, EC, water temperature, and type of filter), with each of the four rivers as a random effect. In the models, the interaction between turbidity and the type of filter paper was considered based on a previous report that examined the effect of suspended solids on eDNA particle distribution size (Barnes et al., 2021). Prior to modeling, the collinearity between environmental factors was determined using generalized variance inflation factors (GVIF) using the R package “car” (Fox & Weisberg, 2019). The factor with the highest adjustment value of GVIFˆ(1/(2*Df)) was repeatedly omitted until all fell below 2 (Fox & Monette, 1992). Through this procedure, the percentages of the forest, farmland, and urban areas and water temperature were excluded from the model. The “glmmTMB” package (Brooks et al., 2017) was used to build the GLMMs using log-transformed eDNA concentration as the dependent variable to identify factors affecting eDNA content. One was added to eDNA concentration prior to log-transformation to accommodate values of 0 (Everts et al., 2024). All statistical analyses were conducted in R ver. 4.3.2 (R Core Team, 2023). Additionally, two generalized linear models (GLMs, gaussian family) were used to evaluate the relationships between filtration time and turbidity and filter paper type, and the relationships between filtration volume, turbidity, and filter paper type. To determine whether the concentration of eDNA differed between small filter and larger filter, we used the paired Wilcoxon rank-sum test.

Results

Measurements of basic water quality parameters

Table 1 shows the measurements of the basic water quality parameters in each river. The ranges of water temperature at each river are as follows: Mizunashi, 14.90 °C–18.00 °C (average, 17.30 ± 0.60); Oka, 18.40 °C–21.60 °C (average, 20.32 ± 0.57); Tagami, 19.00 °C–24.20 °C (average, 21.30 ± 1.08). The water temperature of irrigation canal was recorded only once in November 2022, and it was 19.00 °C. The pH values at each river are: Mizunashi, 6.61–7.96 (average, 7.54 ± 0.48); Oka, 6.09–7.70 (average, 7.18 ± 0.63); Tagami, 5.98–7.27 (average, 6.95 ± 0.49); irrigation canal, 7.08–7.33 (average, 7.20 ± 0.12). Based on observations performed twice daily, the pH values of the samples at the same site tended to decrease over time. The EC values at each river are as follows: Mizunashi, 53.60–119.30 μS/cm (average, 83.40 ± 29.69); Oka, 55.70–76.70 μS/cm (average, 67.70 ± 8.91); Tagami, 39.80–126.60 μS/cm (average, 85.96 ± 36.52); irrigation canal, 25.80–166.20 μS/cm (average, 118.90 ± 80.63). The fluctuations in EC in Tagami and irrigation canal were larger than those in Oka and Mizunashi. The turbidity at each river is as follows: Mizunashi, 5.78–285.82 ppm (average, 78.13 ± 97.69); Oka, 46.73–314.34 ppm (average, 142.23 ± 108.30); Tagami, 4.07–138.54 ppm (average, 31.04 ± 52.86); irrigation canal, 0.45–189.94 ppm (average, 63.78 ± 109.26). Turbidity fluctuated greatly in all sites.

Table 1 Results of water quality analysis.

The “Sampling1” and “Sampling2” indicate the first and second sampling of the day, respectively. Sampling2 was conducted 2 hours after the start of Sampling1. “–” indicates not recorded.

Sample	Turbidity
(ppm)	Electrical conductivity
(μS/cm)	pH	Water temperature (°C)	Sampling time	
2021_9_Tagami_Sampling1	–	–	–	–	15:00	
2021_9_Tagami_Sampling2	–	–	–	–	17:00	
2021_9_Irrigation_Sampling1	–	–	–	–	15:15	
2021_9_Irrigation_Sampling2	–	–	–	–	17:15	
2021_11_Mizunashi_Sampling1	55.2	111.60	7.37	–	10:00	
2021_11_Mizunashi_Sampling2	5.78	119.30	7.37	–	12:00	
2021_11_Tagami_Sampling1	6.72	119.40	7.25	–	11:40	
2021_11_Tagami_Sampling2	4.07	126.60	7.27	–	13:40	
2021_11_Irrigation_Sampling1	0.96	164.70	7.18	–	11:15	
2021_11_Irrigation_Sampling2	0.45	166.20	7.08	–	13:15	
2022_5_Mizunashi_Sampling1	19.30	53.60	7.96	14.90	13:23	
2022_5_Mizunashi_Sampling2	25.38	61.10	7.91	17.70	15:28	
2022_5_Oka_Sampling1	174.52	76.70	7.44	19.70	13:47	
2022_5_Oka_Sampling2	46.73	71.30	7.18	18.40	15:46	
2022_9_Oka_Sampling1	63.25	55.70	7.70	21.60	7:36	
2022_9_Tagami_Sampling1	16.72	67.30	7.11	24.20	8:20	
2022_11_Mizunashi_Sampling1	45.12	98.20	7.87	18.00	7:30	
2022_11_Tagami_Sampling1	138.54	39.80	7.18	19.00	8:40	
2022_11_Irrigation_Sampling1	189.94	25.80	7.33	19.00	8:21	
2023_6_Mizunashi_Sampling1	285.82	–	7.72	17.90	7:29	
2023_6_Mizunashi_Sampling2	110.31	56.60	6.61	18.00	9:54	
2023_6_Oka_Sampling1	314.34	–	7.47	20.90	7:51	
2023_6_Oka_Sampling2	112.33	67.10	6.09	21.00	10:14	
2023_6_Tagami_Sampling1	12.77	–	6.90	21.00	8:42	
2023_6_Tagami_Sampling2	7.43	76.70	5.98	21.00	11:01	

Filtration volume and time

The filtration volume and time for each DNA concentration method are shown in Appendix 3. A higher filtration volume and a shorter filtration time were observed for the large filter than for the small filter. For the samples collected from November 2021 to June 2023, two filter papers were used in 87.71% of the small filter-based filtrations. Among them, the minimum turbidity detected was 4.07 ppm. Additionally, for samples with turbidity equal to or greater than 25.38 ppm, even two filter papers did not allow full filtration (61.10% of the samples). The sample collected during the survey conducted in November 2022 had the longest filtration time (filtration volume: 690 mL, filtration time: 2,610 s). For large filter-based filtration, two filter papers were used for 42.86% of the samples. For samples with turbidity equal to or greater than 110 ppm, a water volume of one L could not be filtered even using two filter papers (72.22% of the samples). Overall, large filters were more efficient in filtering high-turbidity samples than small. Table 2 shows the relationships between filtration volume, filtration time, filter type, and turbidity. The GLMs indicated that turbidity had a significant positive effect on filtration time, and that filter type had a significantly greater effect on increasing filtration time. In addition, turbidity had a significant negative effect on filtration volume, and filter had a significantly negative effect on filtration volume.

Table 2 Effects of filter type and turbidity on filtration time and filtration volume based on the generalized linear models.

(A)					
Time					
Factors	Estimate	Std. Error	z value	p	
(Intercept)	177.72	68.13	2.61	0.009	
Filter	540.52	82.59	6.55	<0.001	
Turbidity	1.82	0.45	4.05	<0.001	
(B)					
Volume					
Factors	Estimate	Std. Error	z value	p	
(Intercept)	1,045.00	24.63	42.45	<2e−16	
Filter	−96.33	29.85	−3.23	0.001	
Turbidity	−2.49	0.17	−14.88	<2e−16	
Notes.

a: filter type and turbidity on filtration time.

b: filter type and turbidity on filtration volume.

Bold font indicates p < 0.05.

eDNA quantification

Target DNA was not detected in the autoclaved distilled water sample transported to the site on each survey date and the PCR control, confirming the absence of contamination between samples during the surveys and reagent contamination. Figure 4 shows the results of the eDNA survey conducted in the rainfall and non-rainfall events. In non-rainfall events, the target eDNA was not detected in most samples, and in the few where it was detected, the concentration was extremely low (0–338 copies/L). In contrast, considerably more target DNA was observed during the rainfall period (November 2022 samples).

Figure 4 EDNA concentrations.

eDNA concentrations during different rainfall events (A) and non-rainfall events (B) “Sampling1” and “Sampling2” indicate the first and second sampling of the day, respectively. Sampling2 was conducted 2 h after the start of Sampling1. “M”, “O”, “T”, and “I” represent the Mizunashi River, Oka River Tagami River and irrigation canal. “F” indicates results of a glass fiber filter with a pore size of 0.7 µm, while “D” indicates a glass fiber filter with a pore size of 2.7 µm. “N.D.” indicates that the target eDNA was not detected. “W ” indicates results of water samples, “S-N ” represents the sediment extraction results based on the method referenced from Nevers et al. (2020) “S-T” represents the sediment extraction results based on the method referenced from Taberlet et al. (2012b).

Figure 4 shows the results of eDNA quantification during the rainfall period. Environmental DNA was detected in 42 out of 47 samples. In terms of survey periods, the eDNA concentrations were in the range of 855–8,267 copies/L for September 2021, 0–1,350 copies/L for November 2021, 250–3,801 copies/L for May 2022, 0–2,730 copies/L for September 2022, 4,807–399,185 copies/L for November 2022, and 0–1,040 copies/L for June 2023. In addition, during the heavy rainfall event of November 29, 2022, high concentrations of cattle DNA were detected, whereas in November 2021, which was a less rainy month, many samples showed low eDNA concentrations or even none. Among the five samples with no eDNA traces collected throughout the entire investigation period, three were from the Mizunashi River. During the survey period on the same date, higher eDNA concentrations tended to be detected in the irrigation channel than in the Tagami River.

Relationships between environmental factors and eDNA concentration

The GLMMs suggested significant positive effects of σ, turbidity, pH, and filter and significant negative effects of waterways distance on eDNA concentration. However, no significant effects of μ, land distance, EC, or the interaction between turbidity and filter were observed (Table 3).

Table 3 GLMM result.

Effects of environmental factors on eDNA concentration based on generalized linear mixed models. “σ” and “μ” represent the variance and mean of normal distribution, indicating duration of rainfall and intense rainfall timing, respectively. “Turbidity: Filter” indicates their interaction term.

Factors	Estimate	Std. Error	z value	p	
(Intercept)	−9.051	3.241	−2.792	0.005	
μ	0.073	0.066	1.106	0.269	
σ	0.533	0.192	2.771	0.006	
Turbidity	0.011	0.005	2.345	0.019	
Land distance	0.003	0.003	1.076	0.282	
Waterways distance	0.009	0.004	−2.194	0.028	
Electrical conductivity	−0.001	0.005	−0.140	0.888	
pH	1.392	0.425	3.273	0.001	
Filter	0.988	0.347	2.842	0.004	
Turbidity: Filter	−0.005	0.004	−1.262	0.207	
Notes.

Bold font indicates p < 0.05.

Discussion

In this study, we investigated the feasibility of detecting low abundance terrestrial mammals that cannot enter the water by analyzing river water during rainfall using glass filter papers with two pore sizes. In addition, we examined environmental factors that may affect the detection of terrestrial mammals. The DNA of the target terrestrial mammals was detected in 42 out of the 47 samples collected during different rainfall events. Thus, we conclude that the terrestrial target species can be consistently detected and quantified from river water samples during rainfall events throughout our surveys. While the target DNA was indeed detected in partial samples during non-rainfall events, its concentration was extremely low. Although some eDNA concentrations detected during the rainfall period were as low as those recorded during the no rainfall period, such intensive rainfall increases streamflow that dilutes eDNA, resulting in extremely low eDNA concentration (or no detection). Therefore, the detection of high concentrations in rainfall samples suggests that the eDNA originated from terrestrial sources.

Previous research indicates that larger pore sizes enhance eDNA capture rates by filter papers used in turbid waters (Barnes et al., 2021). However, in our experiment, the small filter (GF/F) tended to recover more eDNA (p = 0.008), except for four samples with high turbidity, where the large filter (GF/D) recovered more eDNA. Moreover, filter clogging remains an issue when filtering turbid water. Despite the filter GF/D having a larger pore size, excessive turbidity could still prevent it from completely filtering the one L water sample in a short time.

Among all the samples, the highest number of non-detections occurred in the Mizunashi River, involving both small and large filter papers, specifically during the November 2021 sampling period. This phenomenon may be attributed to the relatively weak and short-duration rainfall on the sampling day, which may have limited the transport of eDNA into the water. In addition, the long distance of waterways could have led to eDNA degradation during transport (Joseph et al., 2022). Furthermore, the predominance of forest land in the Mizunashi River area may reduce the intensity of surface runoff (Rahman et al., 2023) because forests generally promote evapotranspiration and rainwater infiltration, resulting in decreased surface runoff. However, because the strong multicollinearity among land uses variables decreased the model accuracy, these variables were not included. Thus, to understand effects of land use, it is fruitful if future research designs eDNA runoff observation in experimental fields with different land uses.

According to the GLMMs results, rainfall duration had a significant positive effect on eDNA concentration, whereas the timing of rainfall peaks did not. Therefore, the amount of rainfall is more important than its intensity (Valentin et al., 2021). In the presence of sufficient rainfall, terrestrial eDNA is continuously transported into rivers, potentially compensating the amount of genetic traces that are lost due to dilution. The observed tendency toward higher eDNA concentrations under highly turbid conditions appeared to be caused by the correlation between turbidity and rainfall with rainfall potentially being the true driving factor behind the increase in eDNA concentration. Terrestrial materials such as soil, organic matter, and terrestrial eDNA are washed into rivers by surface runoff during rainfall, increasing river turbidity. Since eDNA may be sticky (Barnes et al., 2021), it tends to attach to these materials, increasing its particle size and making it more easily captured by filter paper. Similarly, pH was shown to have a significantly positive effect on the eDNA concentration. Therefore, high pH favored the detection of eDNA because low pH tends to degrade eDNA (Jo et al., 2022).

Environmental DNA was consistently detected in the irrigation channel, with no cases of non-detection. This can be explained by the proximity of the irrigation channel to the source point, i.e., at a land distance of only about 17 m. Nonetheless, our GLMMs did not detect any significant association between the land distance and the eDNA concentration. This was likely due to the multiplicity and complexity of the pathways of eDNA transport from land to rivers, which makes it a particularly challenging research topic, whereas we only used the shortest straight-line distance. Although the land distance of the Oka River was only 36 m, the presence of an embankment likely affected the transport of eDNA from land to river. Whether during rainfall or non-rainfall periods, the concentration of eDNA in the irrigation channel was higher than that in the Tagami River. One reason for this is the longer distance of waterways in the Tagami River, along which eDNA undergoes microbial decomposition and is subjected to various physical processes (e.g., sedimentation). In addition, the confluence of tributaries in the Tagami River can contribute to the dilution of eDNA.

Our findings indicate that analyzing eDNA from river water samples collected during rainfall events can enhance the detection of terrestrial mammals. To date, a variety of sampling strategies and substrates have been employed to detect terrestrial vertebrates using eDNA, among which water is the most commonly collected substrate. Compared to aquatic systems, large-scale application of eDNA monitoring in terrestrial ecosystems remains limited, primarily due to the diverse and unstable nature of eDNA sources in these environments (Deiner et al., 2017). Terrestrial eDNA can enter the environment through various trace carriers, such as soil, spider webs, air, pollen, and carrion-feeding insects (Broadhurst et al., 2025), but the applicability of these substrates is often limited. For example, wildflower samples are more suitable for detecting arthropods and birds but not mammals (Thomsen & Sigsgaard, 2019). Spider web eDNA has mainly been used for monitoring invertebrates (Gregorič et al., 2022), and its effectiveness in reflecting vertebrate diversity under natural conditions has not yet been confirmed (Newton et al., 2024). Airborne eDNA has also been used to detect vertebrates, but the results vary substantially across environments—for instance, more mammal species than birds have been detected in zoo settings (Lynggaard et al., 2022), while the opposite pattern is observed in natural environments (Lynggaard et al., 2024). In addition, collecting representative terrestrial eDNA samples is often time-consuming and costly (Gogarten et al., 2020; Seeber & Epp, 2022). In contrast, rainfall events generate surface runoff that washes eDNA from soil, organic debris, and other carriers into rivers, making it possible to monitor many terrestrial species that are otherwise difficult to sample directly from water. This significantly enhances the scope and efficiency of monitoring terrestrial vertebrates using aquatic systems.

However, during rainfall events, the increase in water flow and velocity also had an impact on eDNA concentration and transport (Curtis et al., 2021; Nukazawa, Hamasuna & Suzuki, 2018). Thus, future research based on modeling should focus on more diverse communities, different watersheds and encompass various hydrological factors. Our rainfall data comprised the precipitation amount of 24 h preceding the sampling. However, within each area, the transport time of terrestrial eDNA may vary, resulting in variable sampling times among the sites. This problem can be addressed by hydrologic analysis because it can calculate the eDNA transport time from the source to the river. Furthermore, eDNA was detected in some samples after rainfall events, suggesting that terrestrial eDNA is transported to rivers continuously. Nevertheless, greater flow rates are associated with high turbidity, which increases the risk of safe sampling. Thus, future studies should combine our approach with unmanned sampling technology (e.g., drone) that can collect samples during various rainfall-related events (O’Mahony et al., 2024; Preston et al., 2024).

Conclusion

This study targeted a domesticated terrestrial species that does not directly release eDNA into water bodies to verify the detectability of terrestrial eDNA during rainfall events and explore key environmental factors that affect detectability. We demonstrated that the target terrestrial DNA was successfully detected during weak to strong rainfall events, which highlights the advantage of our approach to monitor terrestrial species from stream water samples. These results suggest that utilizing terrestrial eDNA transported to river water can expand the taxonomic coverage of eDNA monitoring from a sole medium (i.e., stream water) and improve the effectiveness of biodiversity surveillance in a given watershed.

Supplemental Information

Supplemental Information 1 The results of fitting rainfall characteristics with a normal distribution

The “Sampling1” and “Sampling2” indicate the first and second sampling of the day, respectively. The bar chart represents the rainfall amounts, while the red curve represents the normal distribution. “σ” and “μ” represent the parameters of normal distribution. The RMSE value represents the root mean square error between the observed rainfall and the calculated rainfa ll.

Supplemental Information 2 The results of the digital PCR condition investigation

Blue points represent positive wells, while orange points represent negative wells.

Supplemental Information 3 The filtration volume and time of each filter paper, and total filtration volume, filtration time

The “Sampling1” and “Sampling2” indicate the first and second sampling of the day, respectively. “M”, “O”, “T”, and “I” represent the Mizunashi River, Oka River, Tagami River, and irrigation canal. “F” indicates results of a glass fiber filter with a pore size of 0.7 μm while “D” indicates a glass fiber filter with a pore size of 2.7 μm. “R1” and “R2” indicate replicate1 and replicate2. The samples from September 2021 only recorded total filtration volume and filtration time.

Supplemental Information 4 Data integration used in this study

Supplemental Information 5 Quant Studio 3D Digital PCR data

PCR raw data

Supplemental Information 6 Qiacuity One Digital PCR data 2024.8

PCR raw data

Supplemental Information 7 Qiacuity One Digital PCR data 2024.12

PCR raw data

Supplemental Information 8 R code

R code used for data analysis

filter type file Relationship between eDNA concentration and filter type

Filtration time and volume file : used to analyze their relationship with filter type

GLMM file: used to analyze the relationship between eDNA concentration and environmental variables

Rainfall file: used for calculating and fitting rainfall characteristics

We would like to thank Mr. Inoue, Mr. Higuchi, and Mr. Nanri (University of Miyazaki) for their help in experiments and field observations. The authors hereby acknowledge that ChatGPT provided assistance with grammar and wording corrections, as well as language polishing, in this manuscript.

Additional Information and Declarations

Competing Interests

Author Contributions

Field Study Permissions

Data Availability

The authors declare there are no competing interests.

Chen Xu performed the experiments, analyzed the data, prepared figures and/or tables, and approved the final draft.

Kei Nukazawa conceived and designed the experiments, performed the experiments, authored or reviewed drafts of the article, and approved the final draft.

The following information was supplied relating to field study approvals (i.e., approving body and any reference numbers):

No field permit was required. Water sampling was conducted in public river areas that are freely accessible and not subject to specific access restrictions under local regulations.

The following information was supplied regarding data availability:

The raw data are available in the Supplemental Files.

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
