# Peer review of "Detection of terrestrial mammals using environmental DNA during heavy rainfall events and associated influencing factors"

_PeerJ, doi:10.7717/peerj.20166_

## Round 0.1 · original submission · Major Revisions

Thank you for submitting to PeerJ. Your manuscript has been reviewed by two experts in the field, and both find value in the research, but have some suggestions to improve the manuscript. Please respond to each comment in your revision and use their clear, thoughtful suggestions to improve the manuscript in your revision. We look forward to reading it!

Reviewer 1 ·

Basic reporting

Pass

Experimental design

Pass

Validity of the findings

Pass

Additional comments

The manuscript entitled “Detection of terrestrial mammals using environmental DNA during heavy rainfall events and associated influencing factors” presents an interesting study on the detection of terrestrial eDNA in aquatic environments via rainfall-driven runoff. The study's focus on a domestic species (Bos taurus) that doesn't directly interact with the water body is a clever way to isolate the rainfall-mediated transport mechanism, and the use of dPCR for absolute quantification is a strong methodological choice. The abstract and introduction effectively establish the research gap and the study's objectives. The discussion also attempts to interpret the results and acknowledge the limitations. It would be great if authors could include data on the limit of detection (LOD) and limit of quantification (LOQ). This information is crucial as it indicates the smallest number of target DNA copies that can be reliably detected and quantified in the water samples, providing essential context for the study's findings.

L19-20: Rephrase the for flow and conciseness.
“However, an efficient sampling design for terrestrial vertebrate eDNA in aquatic environments has not yet been established because DNA is rarely released into these environments.”
L23: "Bos. taurus" should be "Bos taurus"
L28: “stable detection” should be replaced with “consistently detecting”
L41: “Sea” should be replaced with “water”
L49: "flowers surface" should be "flower surfaces" or "the surface of flowers."
L53: Please delete “which was verified.”
L57: "form water bodies" should be "from water bodies".
L61: Please delete “living”
L68: “In the study” should be replaced with “A study”
L94: "crown of trees" should be "tree crowns" or "the crowns of trees."
L122: In “km2” “2” should be superscripted.
L163: “250-mL” should be written as 0.25 L for consistency.
L204: “samples” should be replaced with “filter papers”
L227: “10-2–10-3” “2” should be superscripted.
L234-235: Please mention the concentration of DNA in ng/ul.
L394: "this may why" should be "this may be why".
L397-398: The explanation for the GF/F filters capturing more eDNA than GF/D despite previous research is interesting but needs to be presented with greater clarity and a more robust explanation, especially regarding the particle size and composition.
L399-400: The reasoning for non-detections in the Mizunashi River is plausible but could be more strongly linked to the model's limitations or findings, rather than just an external hypothesis.
L406: "At the same time though" should be replaced with "Nevertheless" or "However"
L406: "different land use" should be "different land uses".
L444: "Termites" should not be capitalized.
L459-460: "more community" should be “more diverse communities”
L465: "some Sampling" should be "some samples".
L467: "safe surveillance" is unclear here. It might mean "safe sampling" or "safe monitoring."
L475: "which highlight advantage" should be "which highlights the advantage".
L479: "biodiversity surveillances" should be "biodiversity surveillance."
Figure 2 legend: “sediment sample s” remove extra space before “s”
Figure 4 legend: In the “Nevers et al.” year is missing.
Table 3 title: “Here Turbidity:Filter indicates” correct it.

·

Basic reporting

This manuscript explores the detection of terrestrial species' eDNA in waterbodies during rainfall events, under the premise that genetic material is washed from land into adjacent aquatic systems (e.g., rivers, lakes). The authors utilize dPCR to quantify the amount of B. taurus eDNA in samples collected during rainfall, as this species is found in cattle farms throughout the study region but does not come into contact with natural water sources. The study convincingly shows that B. taurus eDNA was more frequently detected in river catchments during rainfall, supporting the notion that rain acts as a vector for transporting terrestrial eDNA into aquatic systems. The authors also examine the effects of filter pore size on eDNA recovery from turbid samples. As expected, smaller pore sizes yielded higher eDNA concentrations, while larger pores facilitated greater filtration volumes and faster processing—an important trade-off, especially in high-turbidity conditions. The effects of environmental variability, as well as distance metrics to/from a waterbody or source of animal eDNA, were also modeled as a part of this study. Multiple factors influenced eDNA concentrations across samples including water turbidity and pH, the filter type used, distance to the waterbody, and the longevity of a rainfall event. These results cumulatively support post-rainfall water systems as a source for biomonitoring both aquatic and terrestrial-based animals, which promotes routine sampling efforts tailored to certain locations/times of the season rather than attempting to cover a broad spatiotemporal sampling range.
While I’m very impressed at the sampling efforts, standardized filtration schemes, advanced dPCR methodology, and thoughtful statistical analysis, I do have some problems with almost each section of this paper when it comes to clarity, cohesiveness, and flow. Within the introduction, several generalized claims are made regarding eDNA methods that do not fully reflect the breadth and nuance of the field. The references cited are limited and overlook much of the relevant literature that challenges or complicates some of the manuscript’s assertions. eDNA methods, including sampling, lysis, and extraction, are often highly context-dependent, tailored to the target species, study system, and research objectives. Please see my line-by-line comments for specific examples.
In addition, throughout the manuscript, grammatical corrections and improved sentence flow would enhance readability. For example, there are repeated issues with verb tense (e.g., using "was" instead of "has been") when referencing past studies, as well as inconsistent punctuation and capitalization. I suggest going through the whole paper and reading it out loud a few times to catch phrasing and sentences that sound a bit jumbled or incomplete.

Regarding article structure and raw data, I do not have any comments or issues with presentation. Please see my attached line-by-line feedback for small improvements/suggestions for your figures and tables.

Experimental design

More detail is needed regarding sampling replication. It appears that 1 L samples consisted of four 250 mL replicates. Were these replicates combined before analysis? If so, how? Was there replication at the PCR level as well? Clarifying this is important to assess whether pseudoreplication might affect your dataset and interpretations.

Validity of the findings

Some parts of the discussion appear disjointed or insufficiently contextualized. For instance, introducing a brief background on the role of organic matter in eDNA transport within the introduction would help ground later statements (e.g., lines 391–398 and 415–419). Additionally, the fourth paragraph of the discussion (lines 433–456) would benefit from restructuring to improve coherence and reduce fragmentation. As it stands, the ideas and supporting citations feel somewhat scattered.

All underlying data and code have been provided, are understandable, and organized. Thank you for providing these materials!

Additional comments

Overall, I’m very pleased with the aims, methodology, and potential impact of this study. With some polishing and further explanation of ideas–and likely one more round of revisions–the manuscript will, in my opinion, be acceptable for publication. Please review my line-by-line suggestions below and take the edits into consideration.

---

## Round 0.2 · Minor Revisions

Thank you for submitting a much-improved manuscript. The reviewers agree that it is nearly ready for publication.

Please see the reviewers’ comments and consider making the suggested changes in your revision.

I look forward to reading your revised manuscript.

Reviewer 1 ·

Basic reporting

NA

Experimental design

NA

Validity of the findings

NA

Additional comments

NA

·

Basic reporting

The authors have made substantial improvements to their manuscript and I find that my earlier concerns about clarity, cohesiveness, and flow have largely been addressed. The introduction is now better contextualized with current research and the discussion has been reorganized to improve readability. Overall, the revised version presents the study in a much clearer and compelling way.
At this stage, my remaining comments mainly concern grammar, phrasing, and reducing ambiguity in some sentences - I encourage the authors to review the manuscript with this in mind.

Experimental design

The scientific rationale, methodology, and analyses are sound and the revisions have addressed my earlier concerns.

Validity of the findings

The findings are supported by the data and I do not have any additional substantial concerns.

Additional comments

My line-by-line suggestions (attached PDF document) highlight areas where small refinements could improve clarity.

---

## Round 0.3 · accepted · Accept

Thank you for making the requested changes, I can see that this version of your manuscript is strong, and well written and ready for publication. Congratulations!